# Externally-forced and intrinsic variability of the Mediterranean surface and overturning circulations

Damien Héron<sup>1</sup>, Thierry Penduff<sup>1</sup>, Jean-Michel Brankart<sup>1</sup>, Pierre Brasseur<sup>1</sup>, Samuel Somot<sup>2</sup>, Robin Waldman<sup>2</sup>, and Romain Pennel<sup>3</sup>

**Correspondence:** Damien Héron (damien.heron@univ-grenoble-alpes.fr)

**Abstract.** Part of Mediterranean Sea variability is forced and paced by external drivers (e.g. atmosphere, river runoffs, Atlantic inflow); the other part has a random phase and spontaneously emerges due to chaotic intrinsic variability (CIV). This study quantifies across time scales the imprints of both variability components on the surface and zonal overturning circulations within the basin, from a 39-year 30-member ensemble ocean  $1/12^{\circ}$  simulation. We find that most of SSH variance is intrinsic over 17% of the basin, in particular in the southern Ionian and Levantine basins, and most notably in the Algerian basin where CIV explains 80% of the SSH variance at periods greater than 4 months. In contrast, 75% of its interannual to decadal variance of the North Ionian Gyre circulation is paced by the atmosphere, suggesting an external triggering of the Adriatic-Ionian Bimodal Oscillating System (BiOS) reversal. Other gyres such as Rhodes, Bonifacio, and Alboran shows more balanced contribution of CIV. Fluctuations of the density-coordinate zonal overturning circulation ( $ZOC_{\sigma}$ ) and of associated transports are mostly paced by the forcing over most of the basin. However, CIV tends to explain a larger fraction of these transports variance in the intermediate and bottom layers near deep convection sites, in particular in the Levantine basin where this fraction exceeds 50% between 27 and 30°E at submonthly periods, and 20-30% at periods reaching 20 years locally. This partly random character of the multi-scale Mediterranean variability has consequences for evaluating model simulations, and the design of observation systems targeting the long-term monitoring of the basin.

#### 15 1 Introduction

The large-scale circulation of the Mediterranean Sea may be sketched as a zonal overturning. The shallow limb is characterized by an eastward surface inflow of Atlantic Water entering through the Strait of Gibraltar. These waters become progressively denser as they flow eastward and finally contribute to the formation of Levantine Intermediate Water (LIW), which feeds a westward return flow at intermediate depth. A counterclockwise cell sits below; deep waters mostly formed by wintertime convection in the Gulf of Lions, the Adriatic Sea, and the Aegean Sea sink and partly feed the lower cell and the west-

<sup>&</sup>lt;sup>1</sup>Université Grenoble Alpes, CNRS, INRAE, IRD, Grenoble INP, Institut des Géosciences de l'Environnement (IGE), Grenoble, France

<sup>&</sup>lt;sup>2</sup>Météo-France, CNRS, Univ. Toulouse, CNRM, Toulouse, France

<sup>&</sup>lt;sup>3</sup>Laboratoire de Météorologie Dynamique (LMD)/IPSL, Ecole Polytechnique, Institut Polytechnique de Paris, ENS, Université PSL, Sorbonne Université, CNRS, Palaiseau, France

50

ward intermediate flow, which then exits the basin within the Gibraltar overflow (Demirov and Pinardi, 2007; Pinardi et al., 2015, 2019).

An overall cyclonic circulation to the north of the basin, and a number of persistent horizontal circulation features such as the Alboran Gyres, the Algerian eddies, the North Ionian Gyre (NIG), are superimposed on this meridionally-integrated overturning upper cell (Millot and Taupier-Letage, 2005; Schroeder et al., 2012). Alongside active mesoscale eddies and other modes of multi-scale variability, these multiple circulation features are influenced by the basin's complex geometry and drive fluctuating three-dimensional transport and redistribution of water properties and mass in three dimensions across the basin (Viúdez et al., 1998; Millot et al., 1990; Gačić et al., 2010; Menna et al., 2019).

This time-varying circulation is forced at first order by external drivers, such as the exchanges with the Atlantic Ocean through the Strait of Gibraltar, the atmospheric forcing, and river runoffs (Pinardi and Masetti, 2000; Demirov and Pinardi, 2007). However, recent studies have shown that a substantial part of the Mediterranean variability may arise spontaneously, i.e. from intrinsic fluctuations generated by ocean nonlinearities. Beyond its oceanic origin, this variability also has a random phase and is thus often labeled "chaotic intrinsic variability" (CIV); this phenomenon therefore acts as a source of uncertainty in oceanic hindcasts and forecasts, and question the attribution of oceanic fluctuations to external drivers solely. Benincasa et al. (2024) showed that a significant fraction of the basin-scale subannual SSH variability is intrinsic, and Waldman et al. (2018, 2017) showed that deep convection in the Gulf of Lions is modulated by a strong sub- to inter-annual intrinsic variability. However, these studies were focused on specific regions or relatively short timescales, leaving the low-frequency intrinsic variability partly unexplored over the basin. On the other hand, Rubino et al. (2023) and Gnesotto et al. (2024) showed from idealized eddying simulations of the basin that mesoscale—topography interactions alone can generate interannual-to-decadal intrinsic variability, even without variable forcing. Yet, the extent to which such intrinsic mechanisms may operate in more realistic configurations and compete with external drivers in generating variability remains to be quantified.

The OceaniC Chaos – ImPacts, strUcture, predicTability (OCCIPUT) project has shown from an ensemble of 1/4° global ocean/sea-ice simulations (Bessières et al., 2017; Penduff et al., 2014) that the contribution of CIV may indeed compete with, and locally dominate the (externally-) forced variability of several oceanic indices in many regions of the global ocean, up to the scale of basin gyres and multiple decades. For instance, Leroux et al. (2018) and Carret et al. (2021) emphasized the substantial imprint of CIV on the subannual-to-decadal Atlantic overturning circulation and on regional SSH fields, respectively. However, the relatively modest OCCIPUT model resolution limits its ability to represent fine-scale Mediterranean dynamics, especially mesoscale—topography interactions that are likely important for the generation of intrinsic variability in this basin.

The goal of the present study is to evaluate the imprints of (atmospherically-)forced and intrinsic ocean variability over the Mediterranean Sea, with an emphasis on interannual and longer scales. We use a relatively large (30 members) and high resolution (1/12°) realistic ensemble simulation spanning almost 4 decades to disentangle the external and intrinsic sources of Mediterranean variability, beyond the seasonal timescales on which Benincasa et al. (2024) focused. Compared with OC-CIPUT, this 1/12° configuration is expected to provide a more realistic representation of mesoscale activity, boundary current

instabilities, and dense water formation processes in the Mediterranean Sea.

We focus on the variability of three complementary indices derived from the simulation and from observations when available: (i) SSH, which captures the imprint of mesoscale and horizontal circulation; (ii) SSH-derived relative vorticity of selected gyres and recirculation features; and (iii) components of the zonal overturning circulation (ZOC) computed in density space, which link the surface, intermediate and bottom branches of the Mediterranean large-scale circulation.

The remainder of the paper is organized as follows. Section 2 describes the ensemble simulation, the observational products, and our post-processing techniques. Section 3 provides an assessment of the model skill with respect to observational products and existing studies from subannual to decadal timescales. Section 4 then assesses the contributions of external and intrinsic sources of this multi-scale variability. Section 5 summarizes the main conclusions and discusses the broader implications of these findings.

#### 2 Datasets and methods

## 2.1 Ensemble ocean simulation

Our ensemble ocean simulation is based on the NEMOMED12 configuration of the NEMO ocean model, implemented over the Mediterranean basin at 1/12° resolution and over 75 vertical levels, and driven by specified forcing between 1979 and 2017. Beuvier et al. (2012), Adloff et al. (2015), Hamon et al. (2016), and Waldman et al. (2017) have presented this model configuration in detail, and evaluated its skill in representing the circulation and water masses of the basin, when driven by earlier versions of our atmospheric forcing.

Our surface forcing is provided by the ALDERA3 dataset, which consists in 3-hourly air-sea fluxes (wind stress, heat and freshwater fluxes). This dataset was developed by dynamically downscaling the ERA5 reanalysis at 12 km resolution with the version 6 of the CNRM-ALADIN regional climate model Nabat et al., 2020 using a spectral nudging option Colin et al., 2010 to increase the large-scale control of ALADIN by ERA5. With respect to the first version of ALDERA (Hamon et al., 2016), ALDERA3 was designed to improve the output temporal resolution, cover the more recent years thanks to the ERA5 forcing, improve the large-scale chronology via the spectral nudging technique, use a smoother interpolation of the sea surface temperature (SST) forcing, and add an explicit representation of natural and anthropogenic aerosols interacting with the radiative and cloud schemes (see Nabat et al., 2020). ALDERA3 covers the whole Med-CORDEX domain and is a contribution to this international initiative (Ruti et al., 2016). A Newtonian relaxation is applied to NEMOMED12 SSTs during the simulation to maintain thermal stability, with a restoring coefficient equivalent to  $40 \ W \cdot m^{-2} \cdot K^{-1}$ .

River runoff is prescribed at the river mouths and along the coastline as a surface freshwater flux. 1980-2017 Climatological seasonal cycle derived from the interannual dataset of Ludwig et al. (2009) is used. Major Mediterranean rivers fluxes are directly taken from the database, other smaller rivers are gathered into a coastal runoff (Hamon et al., 2016; Waldman et al., 2017). Boundary forcing in the Gulf of Cadiz is applied in each ensemble member within a buffer zone, where the model temperature, salinity and SSH fields are relaxed towards their monthly climatological counterparts extracted from the ORAS4

global ocean reanalysis (Balmaseda et al., 2013), with a relaxation time scale decreasing westwards. More details about the model configuration and the buffer zone are given in Beuvier et al. (2012); Adloff et al. (2018).

The simulation starts with a 20-year one-member spin-up (1979-1998), which adjusts the upper part of the basin. The 30 ensemble members are then initialized by this spun-up state, and subsequently driven by the same surface and lateral fluxes throughout the 39-year integration period (from January 1979 to December 2017). Ensemble dispersion is generated by introducing slight stochastic perturbations to the biharmonic viscosity coefficient (perturbations amount to 0.01% of the nominal coefficient) during the first 6 months of the run. These tiny perturbations are shut off on June 27th, 1979 for the rest of the integration. The ensemble spread grows and stabilizes in amplitude after about 3 years (not shown).

We will thus focus our analyses over the period 1982-2017, over which all ensemble members share the same model setup, the same atmospheric and lateral forcing, ensuring that ensemble dispersion is solely due to intrinsic ocean variability emerging inside the domain. The use of climatological relaxation fields in the Gulf of Cadiz implies that non-seasonal Atlantic variability is absent from our simulation; in other words, the forced and intrinsic non-seasonal variabilities on which we shall focus are endemic to the Mediterranean basin.

# 2.2 Sea Surface Height observational product

The observational dataset used in this study is the European Seas Gridded Sea Surface Height product, distributed by the Copernicus Marine Service (DOI: https://doi.org/10.48670/moi-00141). This product provides daily sea level fields on a 1/8°×1/8° grid over the European seas, covering the period from January 1993 to December 2017. It provides Absolute Dynamic Topography (ADT) derived by combining Sea Level Anomalies (SLA) with a Mean Dynamic Topography using the DUACS system (Data Unification and Altimeter Combination System). ADT is used as the observational counterpart to the model SSH, allowing direct comparison between modeled and observed daily sea level fields over the Mediterranean Sea, after subtraction of their spatial means.

## 110 2.3 Diagnostics

100

105

## 2.3.1 Geostrophic vorticity

The vertical component of geostrophic relative vorticity is derived from SSH as

$$\zeta = \frac{g}{f} \nabla^2 SSH$$

within each ensemble member and from the observational product, where *g* is the gravitational acceleration and *f* the Coriolis parameter. We computed vorticity from low-frequency SSH fields (see section 2.3.3), and averaged the result over localized boxes shown in Figure 1. The resulting timeseries characterize the slow evolution of regional gyre-like features of interest.

120

## 2.3.2 Overturning

The Mediterranean zonal overturning circulation provides a quantitative meridionally-integrated view of volume transports and water mass transformations within the basin, from their surface inflow to their subsurface outflow through Gibraltar Strait (Sayol et al., 2023). In order to assess the imprint of forced and intrinsic variability on the zonal transports in various density classes throughout the basin, we computed in each ensemble member the zonal overturning circulation ( $ZOC_{\sigma}$ , in Sverdrups:  $10^6m^2.s^{-1}$ ) in density space from potential density referenced at the surface ( $\sigma_0$ ) and velocity fields saved as successive 5-daily averages, as described in Hirschi et al. (2020):

$$ZOC_{\sigma}(x,\sigma_0,t,n) = \int\limits_{y_1}^{y_2} \int\limits_{-h(x,y)}^{\eta} u(x,y,z,t,n) \times H(\sigma_0 - \sigma_0(x,y,z,t,n)) \, dy \, dz$$

where y1 and y2 denote the basin's latitudinal boundaries ( $5^{\circ}N$  and  $45^{\circ}N$ ),  $\eta$  the free surface height, h(x,y) the bottom depth,  $\sigma_0$  the reference isopycnal for transport integration, u(x, y, z, t, n) the zonal velocity at depth z and time t in member n, and H the Heaviside step function with H(x) = 1 for x > 0 and H(x) = 0 otherwise. Time series of zonal transports are finally derived from  $ZOC_{\sigma}$  as explained in section 3.2, post-processed as explained in the next two sections, and analyzed in the rest of the paper.

## 130 2.3.3 Temporal scale decomposition

The 1979-2017 linear trends and mean seasonal cycles are removed from all SSH and ZOC-derived time series before decomposing them into temporal scales. Each of the resulting time series, noted  $X(\mathbf{x},t)$  at location  $\mathbf{x}$  and time t, may be then decomposed following Sérazin et al. (2015):

$$X(\mathbf{x},t) = \overline{X}(\mathbf{x}) + X_{LF}(\mathbf{x},t) + X_{HF}(\mathbf{x},t),$$

135 where

- $-\overline{X}(\mathbf{x})$  is the temporal average;
- $X_{LF}(\mathbf{x},t)$  is the low-frequency (LF) component (variability with periods longer than about 2.5 years) calculated by applying a low-pass Lanczos filter with a cut-off period of 912 days; due to filtering side effects, the first and last 2 years of filtered time series are discarded.
- $X_{\rm HF}({\bf x},t)=X({\bf x},t)-X_{\rm LF}({\bf x},t)$  is the high-frequency (HF) component. Interannual variability is thus included in the HF component.

## 2.3.4 Total, intrinsic and forced variabilities; intrinsic variance ratios

All variance estimates presented below were computed from demeaned, detrended, deseasonalized time series within all members, with or without decomposition into temporal scales (see 2.3.3). Let  $X_n(t)$  be the time series of a variable of interest in

155

- the n-th ensemble member, with  $t \in [1, T]$ ,  $n \in [1, N]$ , and N = 30. Let overbars and brackets denote temporal and ensemble means, respectively:  $\overline{X_n} = \frac{1}{T} \sum_{t=1}^T X_n(t)$ , and  $\langle X \rangle(t) = \frac{1}{N} \sum_{n=1}^N X_n(t)$ . These operators are used to split the total variability  $X_n(t)$  into its (externally-) forced and intrinsic components following Hogg et al. (2022).
  - The forced variability component is common to all members, and is thus estimated as the time-varying ensemble mean:  $X_F(t) = \langle X \rangle(t)$ .
- The intrinsic variability component in member n is computed by subtracting the forced from the total variability:  $X_{I,n}(t) = X_n(t) X_F(t)$ . By construction,  $\overline{X_{I,n}} = 0, \forall n$ .

The variances of the total signal  $X_n$ , and of its forced  $(X_F)$  and intrinsic  $(X_I)$  components are estimated and compared as follows:

- Mean variance of the total variability:  $\sigma_T^2 = \langle \overline{(X_n(t) \overline{X_n})^2} \rangle$
- Mean variance of the forced variability:  $\sigma_F^2 = \overline{(X_F(t) \overline{X_F})^2}$
- Time dependent variance of the intrinsic variability (or instantaneous ensemble variance):  $\sigma_I^2(t) = \langle X_{I,n}(t)^2 \rangle$
- Time dependent intrinsic variance ratio (or instantaneous contribution of intrinsic processes to the total variance):  $R_i(t) = \frac{\sigma_I^2(t)}{\sigma_T^2}$
- Mean variance of the intrinsic variability:  $\sigma_I^2 = \overline{\sigma_I^2(t)}$
- 160 Mean intrinsic variance ratio  $R_i = \overline{R_i(t)}$ .

Our choice of biased variance estimates applied on zero-mean detrended time series ensures that  $\sigma_T^2 = \sigma_F^2 + \sigma_I^2$  (Narinc et al., 2024).

#### 3 Assessment of the ensemble simulation

Before disentangling the forced and intrinsic components of variability from our simulation, we first assess how well the model driven by ALDERA3 reproduces the mean features and the total variability of the Mediterranean horizontal and overturning circulations, at both low- and high-frequency.

## 3.1 Sea Surface Height (SSH)

## 3.1.1 Time averaged SSH

At first order, the Mediterranean surface circulation is characterized by cyclonic and anticyclonic gyres and semi-permanent jets. These coherent structures persist throughout the simulation and are clearly visible in both the time-averaged altimetric observations and the time-ensemble-averaged model fields, shown in Figure 1a-b.

**Figure 1.** Time-averaged sea surface height (SSH) between 1995 and 2015 from the Copernicus observational product (a) and from the ensemble-averaged simulation outputs (b). Total sea surface height (SSH) standard deviation from satellite altimetry at high frequency (c) and low frequency (d). The same quantities derived from the simulation are shown respectively in e) and f). The black boxes surround the gyres investigated in Section 4.2

The model skillfully simulates the time-averaged general circulation in most parts of the basin, in particular large-scale features such as the inflow of Atlantic waters along the north African coast, the cyclonic gyres in the Gulf of Lions, in the Southern Adriatic Sea, and the Rhodes Gyre. The main simulation-observation difference appears in the Balearic sea (2°W to 1°E along the Spanish coast), where the observations reveal a weak cyclonic geostrophic circulation that has an opposite sign

in the simulation. In the south and west of Crete, smaller circulation features —Ierapetra (35°N;26°E) and Pelops (36°N;22°E) gyres — appear less clearly in the model.

#### 3.1.2 Total variability ( $\sigma_T$ ) of SSH

In both the observations and the simulation, the high frequency (HF) total SSH variability reaches its maxima along the Algerian jet in the western basin, in the Ionian Sea, and in the Levantine Sea (figures 1 c,d). The model slightly underestimates the observed HF total variability, as was reported by Beuvier et al. (2012) in a similar but non-ensemblist NEMO simulation; it is also possible in our case that the underestimation of local HF variability maxima is enhanced by the ensemble averaging process that smoothes out inter-member differences. A more local discrepancy is found southeast of Crete, in the Ierapetra gyre, where the simulated HF variability is weaker than observed. Aside from these specific differences, the model reproduces HF SSH variability reasonably well, with standard deviations of 3–4 cm matching observations.

The model correctly reproduces the low-frequency total SSH variability (Figure 1 e,f), capturing the main observed patterns and amplitudes. In particular, the North Ionian Gyre (NIG) stands out as one of the dominant modes of LF variability, both in the observation and the simulation. Observations reveal local LF variability maxima around the Cretan Sea, but their simulated counterparts appear too weak, suggesting a lack of energetic regional variability there. Conversely, the model overestimates the variability in the Balearic Sea (39.5°N;2.5°E), as also reported by Waldman et al. (2018); Hamon et al. (2016) using earlier versions of the same model configuration. Based on these results, we excluded the Ierapetra, Pelops, and Balearic gyres from the subsequent analysis of vorticity in Section 4.2. These gyres either exhibit too small low-frequency variability in the observations, or significant discrepancies in their modeled representation—particularly in terms of amplitude and persistence. The five gyres retained for their subsequent analysis in terms of forced and intrinsic LF fluctuations are those that are correctly located and exhibit a substantial variability in both the simulated and observed datasets.

## 3.2 Zonal Overturning Streamfunction in density coordinates (ZOC)

Since  $ZOC_{\sigma}$  represents the zonal transport cumulated from the densest waters up to a lighter density class, its maximum along the density axis indicates at each longitude the intensity of the clockwise upper overturning cell, and its minimum at depth the intensity of the counterclockwise lower cell (where and when it exists). Depending on longitude, these two cells are associated with two or three opposite flows stacked vertically:

- 1. Surface eastward flow: its transport is given by the difference between the maximum of  $ZOC_{\sigma}$  and its value at the lightest density level (surface).
- 2. Intermediate westward flow: its transport is given by the difference between the  $ZOC_{\sigma}$  minimum in dense classes and its aforementioned maximum.
- 3. Bottom eastward flow: where it exists, its transport is given by  $ZOC_{\sigma}$  in the densest class (zero by construction) and the  $ZOC_{\sigma}$  minimum.

Figure 2. (a) Time-ensemble-averaged  $ZOC_{\sigma}$  as a function of longitude and potential density  $(\sigma_0)$ . Density classes with transports smaller than 0.05 Sv are shown in white. The grey contour indicates -0.05 Sv (dotted line) and 0.05 Sv (solid line). (b) Mean transport of the surface (black), intermediate (blue), and bottom (green) zonal flows associated with the overturning cells, calculated from the differences  $ZOC_{\sigma}(max) - ZOC_{\sigma}(surface)$ ,  $ZOC_{\sigma}(min) - ZOC_{\sigma}(max)$ , and  $ZOC_{\sigma}(bottom) - ZOC_{\sigma}(min)$ , respectively. Min and Max are calculated before the time-ensemble average, and so are not directly comparable with panel a. Transparency indicates where time-mean zonal transports fall below 0.05 Sv. In both figures, the longitude of the Tarifa Narrows is shown as a red line.

# 3.2.1 Time averaged ZOC and zonal transports

210

Figure 2a shows the time-ensemble-mean zonal overturning circulation in density space  $(ZOC_{\sigma})$  in the basin.  $ZOC_{\sigma}$  reaches its maximum around 28  $kg.m^{-3}$  near Gibraltar and around 28.9  $kg.m^{-3}$  around 24°E. This limit marks the center of the clockwise upper overturning cell, and the lower boundary of its upper limb (surface eastward flow). Atlantic Waters (AW)

220

225

crossing the Gibraltar Straits are carried eastwards within this branch, get progressively denser during their journey as evaporation salinizes and cools them. In the eastern basin, most of the surface waters have become dense enough to sink and flow back underneath towards Gibraltar Straits, within the lower limb of this upper cell (intermediate westward flow). Part of these westward-flowing waters get lighter and are upwelled back into the upper limb in the western basin, but most of them cross the Straits and overflow into the Atlantic Ocean.

The simulated transports at Gibraltar are evaluated at the longitude of the Tarifa narrow  $(5.5^{\circ}W)$ , as done by Soto-Navarro et al. (2015). The time-ensemble-mean eastward transport of Atlantic waters there is 0.79Sv, close to the  $0.85\pm0.05Sv$  estimate deduced by combining current meter observations and basin-scale mass balance (Gonzalez et al., 2025). The time-ensemble-mean westward transport of Mediterranean waters underneath is 0.74Sv at the same longitude. This is also close to observed reconstructions, with an estimated outflow of  $0.81\pm0.05Sv$  from INGRES measurements (Sammartino et al., 2015)

The time-ensemble-mean of the surface eastward transport (Figure 2b) increases from about 0.74 Sv at Tarifa Narrows (5.5°W) to 1.2 Sv near 6°E (Gulf of Lions and Algerian basin), mostly due to the aforementioned upwelling of denser waters in the western basin (Waldman et al., 2018). This transport smoothly decreases between 6°E and 15°E and remains close to 1 Sv up to 27°E; it decreases down to zero in the easternmost part of the basin as this 1 Sv of AW sinks, ultimately feeding the westward-flowing intermediate layer. This latter transport shows overall consistent zonal variations with the surface eastward transport, the bottom transport being largely second-order west of 25°E.

The time-ensemble-mean  $ZOC_{\sigma}$  upper cell structure and amplitude are qualitatively—if not quantitatively—consistent with those described by Sayol et al. (2023) based on a 1/12° reanalysis. Their Figure 3c shows a similar basin-wide upper cell in density coordinates, with a similar zonal profile of the surface eastward transport. Our ensemble captures the longitudinal alternation of small transport maxima (associated with secondary overturning cells), with peaks located around 5°E and 11°E in the western basin, and 17°, 18.5°, and 24°E in the eastern Mediterranean—slightly shifted eastward relative to the reanalysis. Another difference is found in the maximum density of the surface eastward flow in the western basin, which is about 0.3-0.4  $kg.m^{-3}$  smaller in our simulation than in the reanalysis, likely reflecting some differences in buoyancy forcing, convection patterns or water mass properties. Overall, the model shows good agreement with the representation of Atlantic inflow into the Mediterranean, the eastward transport of AW throughout its longitudinal range, and the rate and location of AW sinking throughout the basin.

Slightly negative  $ZOC_{\sigma}$  extrema are found at largest densities at certain deep locations east of 3°E; they correspond to the counterclockwise bottom overturning cell, which feeds the bottom eastward flow, and locally intensifies the westward intermediate flow. This lower cell, which is very likely influenced by dense water formation in the Gulf of Lions, Aegean sea and Adriatic sea, is weaker than the upper cell: its lower limb drives an eastward mean bottom transport of about 0.1 Sv around 4°E, 0.2 Sv at 19°E and 0.34 Sv at 29°E. Its upper limb slightly intensifies the intermediate westward transport with the same time-mean amplitude, thus explaining the slight asymmetry between the black and blue lines in Figure 2b.

**Figure 3.** Standard deviation of the total variability of surface (black), intermediate (blue), and bottom (green) zonal transports at high frequency (a) and low frequency (b). Transparency indicates where time-mean zonal transports fall below 0.05 Sv.

# 3.2.2 Total variability $(\sigma_T)$ of zonal transports

The high-frequency standard deviation of the surface eastward and intermediate westward transports (Figure 3 a, blue and black lines) is of the same order of magnitude as their time-mean values (standard deviation of about 0.4 Sv and a mean value of about 1 Sv), indicating substantial HF fluctuations in the upper *ZOC* cell. In the surface eastward branch, the HF standard deviation reaches its maximum (0.55 Sv) at 5°E, i.e. near the Gulf of Lions deep convection site, and remains close to 0.4 Sv from 19 to 31°E. Its low-frequency counterpart (Figure 3 b, black) is 3-5 times smaller, and exhibits an opposite zonal asymmetry with larger values in the eastern basin ( up to 0.14 Sv at 23°E). This suggests that the atmospheric forcing and/or intrinsic ocean dynamics drive *ZOC* fluctuations with different time scales in both halves of the basin. This hypothesis is addressed in the next section, where total variability is split into forced and intrinsic components.

255

260

The total HF variability of the intermediate westward transport is about 20% weaker than that of the eastward surface transport west of 10°E. This difference may be associated with the local maximum of HF variability of bottom eastward transport at 4°E. There, the transport can reach nearly three times its mean value, likely due to short and intense deep-water formation events in the Gulf of Lions.

West of 10°E, the total HF variability of intermediate westward transports is smaller than that of surface eastward transport, suggesting that short-term anomalies of the upper and lower ZOC cells have the same sign (both clockwise or both counterclockwise). In the eastern basin, LF variability shows the opposite behaviour: intermediate transport fluctuations dominate over those at the surface, especially near 19°E and 28°E, close to the Adriatic and Aegean deep convection sites. This suggests that long-term anomalies in the upper and lower ZOC cells often tend to have opposite signs, i.e. one clockwise and the other counterclockwise. Although the origin of these east—west differences lies beyond the scope of this study, they raise the hypothesis that deep convection impacts zonal transport anomalies differently across the three isopycnal layers.

## 4 Contributions of forced and intrinsic variability

We have described at this point the total variability of our variables of interest in the simulation in comparison with available observations. We now take advantage of ensemble statistics introduced in section 2.3.4 to split the simulated variability at various time scales into its forced and intrinsic components, and compare them.

# 4.1 Sea Surface Height

#### 4.1.1 High-frequency SSH variability

The left and middle panels in Figure 4 show the decomposition of the SSH variability into its forced and intrinsic components, at HF (top) and at LF (bottom).

The forced high-frequency SSH variability (Figure 4a) is more pronounced in the western than in the eastern basin (1 cm of variability difference). Local maxima are visible in several coastal regions, potentially driven by strong HF variability of dominant winds (Tramontane and Mistral in the Gulf of Lions, Meltemi in the northern Aegean Sea, Bora in the northern Adriatic). In the southern part of the basin, substantial forced HF variability is also found in the Alboran Sea and in the Gulf of Gabès near the Tunisian shelf. In the same region, intrinsic HF variability exhibits a different spatial distribution, likely associated with eddy-active regions along the Algerian coast, the Ionian Basin and in the Levantine Basin (Figure 4b). The strong intrinsic HF signal along the Algerian coast likely reflects the combined effect of baroclinic instability and the interaction of currents with complex coastal topography (Demirov and Pinardi, 2007; Puillat et al., 2002). These processes can lead to the spontaneous generation of energetic mesoscale structures whose high-frequency random fluctuations are relatively independent of direct atmospheric fluctuations. A marked intrinsic signal is also found near 29°E-36°N, within the northern part of the Rhodes Gyre.

285

295

Figure 4. Forced and intrinsic standard deviations of the HF SSH variability (panels a and b) and of the LF SSH variability (panels d and e). The right column shows corresponding mean intrinsic variance ratios  $R_i$  defined in Section 2.3.4, at HF and LF (panels c and f); the black contour corresponds to  $R_i = 0.5$ .

Intrinsic HF variability remains negligible in shallow regions such as the Adriatic and Aegean Seas, the Gulf of Lions, and over the Tunisian shelf; the model resolution may not be fine enough there to resolve hydrodynamic instabilities because of small Rossby radii.

The HF intrinsic variance ratio is shown in Figure 4c A ratio of 1 indicates that variability is entirely intrinsic, and a value of 0 reflects purely forced variability. Intrinsic processes account for about 20% of the HF SSH variance over the shallow coastal areas (Aegean Sea, Adriatic Sea, Tunisian shelf), and about 40-50% in the central Mediterranean. The largest HF ratios are found along the Algerian coast (up to 60%), in the Levantine Basin and in the South Ionian sea.

## 290 4.1.2 Low-frequency SSH variability

Maxima in forced LF variability (up to 7 cm) are found in the Balearic, Alboran and Ionian Seas (Figure 4d). In these regions, intrinsic LF variability is about 2-3 times smaller (2-3 cm, Figure 4e), indicating that low-frequency SSH anomalies there are mostly driven by the atmosphere. This intrinsic LF component exhibits a more homogeneous distribution across the central and southern Mediterranean, with notable maxima east of the Alboran Sea and within the Algerian Basin, and remains small in most shallow regions as reported at HF.

However, the intrinsic component explains more than 50% of the LF SSH variance in several regions, including the Algerian Basin, the Tyrrhenian Sea, the southern Ionian Sea, and the Levantine Basin. Altogether, these areas represent approximately 17% of the total basin area both at HF and LF, and are located in the same regions at both time scales with two exceptions:  $R_i$  exceeds 50% at LF and not at HF in the Tyrrhenian Sea, and the opposite is found in the central Ionian Sea. Considerable LF intrinsic variance ratios are found along the Algerian coast, where up to 75% of the LF SSH variance is of intrinsic origin.

305

310

Small intrinsic variance ratios characterize shallow regions, where the LF variability is thus mostly paced by the atmospheric forcing.

## 4.1.3 Spectral analysis of SSH variability

The variance maps presented above provide a spatial view of SSH variability in two wide classes of time scales. We now extend our analysis to the entire frequency spectrum by computing the power spectral density (PSD) of SSH variability components between 2 days and 18 years, focusing on two latitudes where  $R_i$  is largest:  $38^{\circ}$ N in the western basin and  $34^{\circ}$ N in the eastern basin (Figure 4c,f). Colors in figures 5a and 5b show along this line the longitude-dependent PSDs of the total and intrinsic variability, respectively.

The total and intrinsic SSH variability spectra are red throughout the basin. Consistently with Figures 1d and 1f, the total spectrum shows moderate longitudinal contrasts at subannual periods at both latitudes, and localized peaks at periods longer than 1-2 years. The second spectrum reveals that high-frequency intrinsic variability (0.2-0.5 year periods) is much larger in the western basin where the 38°N line intersects the eddy-active Algerian current. More generally, the SSH intrinsic variability is strongest at periods longer than 1 year along both latitudes, except between 11 and 15°E above shallow topography south of Sicily.

The isolines in Figure 5 represent the intrinsic-to-total PSD ratio. As for  $R_i$ , a ratio equal to 1 indicates that the variability at a given frequency and longitude is totally intrinsic, while a value of 0 corresponds to purely forced variability. More than 75% of the SSH variability is paced by the atmosphere outside the thin black line ( $R_i 

Figure 5. Colors show the Power Spectral Density (PSD) of total (a) and intrinsic (b) SSH variability as a function of longitude along the zonal lines shown in Figure 4c. PSDs were computed within each member from 38-year deseasonalized and detrended daily SSH timeseries, without (a) and with (b) prior removal of their ensemble mean, and then ensemble averaged. PSDs are shown using a  $\log 10$  scale  $(cm^2.day)$ . The intrinsic-to-total PSD ratio is shown in both panels as 25%, 50%, and 75% isolines (thin solid, thick black, and red contours, respectively). To improve readability, both PSDs were smoothed in the frequency domain using a 11-point Hanning window as done in Leroux et al. (2018).

Algerian gyre around 1998 where the members shown in blue and green in the left panel generate opposite vorticity anomalies.

The different phases of these gyres in various members clearly illustrates the advantage of ensemble over single simulations for attributing specific events to external drivers (also see Narinc et al., 2024).

Figure 6. (a) Low frequency evolution of geostrophic vorticity anomalies within the boxes shown in Figure 1, computed as explained in Section 4.2. Two randomly selected members are shown in blue and green, the remaining 28 ensemble members are shown in grey, and the vorticity derived from AVISO is shown in orange. (b) Temporal evolution of the intrinsic variance ratio  $R_i(t)$  for each gyre. The blue line indicates the 0.5 threshold, above which intrinsic variability dominates.

350

365

**Eastern Alboran Gyre:** the upper panels of Figure 6 confirm the substantial imprint of intrinsic variability in this region, as already reported by Peliz et al. (2013). The intrinsic variance ratio \$R\_i(t)\$ exhibits strong temporal changes in this gyre: its LF variability is mostly forced over certain periods (e.g., 1990-1997) and intrinsic at other times (e.g., 1997-2008, 2011 and 2014). Hints of bimodal intrinsic behaviors also seem to appear in 2005–2008, when 5 members simulate positive vorticity anomalies, most others simulate the opposite, and a few switched modes.

Algerian Gyre: in agreement with SSH diagnostics, the Algerian gyre LF vorticity fluctuations match AVISO in terms of amplitude, and remain strongly random during these 35 years ( $65\% 

375

380

Figure 7. Forced (left) and intrinsic (middle) variabilities in Sv of the surface (black), intermediate (blue), and bottom (green) zonal transports at high (first line) and low (second line) frequency. Corresponding mean intrinsic variance ratios ( $R_i$ ) are shown in the right panels. Color Transparency indicates where time-mean zonal transports fall below 0.05 Sv.

# 4.3.1 High-frequency variability of zonal transports

The forced and total HF variability of the surface eastward transport closely match each other (black lines in Figures 3a and 7a), indicating the atmospheric pacing of most of its variance. Intrinsic variability remains modest ( $\sim$ 0.1 Sv, Figure 7b), keeping  $R_i$  around 10% throughout most of the basin. Similar values of  $R_i$  are found in the intermediate westward transport, except near 29°E where HF intrinsic fluctuations peak and raise  $R_i$  up to 37% (blue lines in Figures 7b–c).

In the western basin, forced HF fluctuations are slightly stronger at the surface than at the intermediate level (by about 0.05 Sv, Figure 7a), suggesting positively-correlated, atmospherically-paced variations of the upper and lower ZOC cells. Intrinsic HF fluctuations exhibit the opposite pattern in the eastern basin, being larger at the intermediate level (Figure 7b), which suggests anti-correlated random variations of the two cells (see Section 3.2.2).

The strongest HF intrinsic variance ratio is found in the bottom eastward transport (green line in Figure 7c):  $R_i$  at 6°E and 19°E is 2–3 times larger than in the upper branches, and reaches 50% at 29°E. These localized maxima incidentally coincide with deep convection sites (Gulf of Lions, Adriatic and eastern Levantine basins), raising the hypothesis that the ensemble spread associated with short-term convection events (see Waldman et al., 2017) propagates downward and induce random HF transport fluctuations. Testing this hypothesis is left for future studies.

# 385 4.3.2 Low-frequency variability of zonal transports

Figure 7d–e shows that the forced and intrinsic LF variabilities of surface and intermediate transports are largest between 20 and 32°E, with more uniform and smaller amplitudes in the western basin. Forced fluctuations of both transports are smallest west of 1-2°W, probably since no LF variability is forced into the basin through the western buffer zone; this probably explains

why intrinsic variance ratios  $R_i$  in the uppermost 2 layers reach a local maximum (about 30%, black and blue lines in Figure 7f) in the Alboran basin. Values of  $R_i$  decrease further east and get down to 10% near 3°E, progressively increase again toward the east and reach another peak near 35°E.

The forced LF variability of bottom transports (green line in Figure 7d) is maximum in the central and eastern basins, unlike its HF counterpart. In contrast, the LF intrinsic variability of bottom transports has the same zonal distribution as its HF counterpart with a 3-4 time smaller standard deviation. As at HF, the  $R_i$  of bottom transports at LF exceeds that of the other two layers east of 25°E, where 15 to 50% of the bottom transport variance has a random phase.

To summarize, the contribution of intrinsic processes on the HF and LF variance of zonal transports is globally largest in the eastern basin and in dense layers, with an additional maximum west of 3°E in the lightest two layers. More than 30-40% of the variance of such transports can be random in phase locally, advocating for a more thorough analysis of their underlying dynamics, and for the need to keep in mind the importance of this intrinsic driver of random ZOC variability that competes with the atmospheric one.

## 4.3.3 Spectral analysis of intermediate transport variability

Figure 8a shows over the whole longitudinal and frequency domains the total variability of the intermediate westward transport. In accordance with Figure 3a, the total variability PSD exhibits localized minima at the eastern end of the domain, around 26°E where topography is shallow, and at its western boundary where the lateral forcing is purely seasonal and intrinsic variability is damped. The intrinsic variability of this intermediate transport (Figure 8b) reaches two maxima at LF in the eastern basin, in particular in the eastern Levantine basin, close to the Aegan convective site (26°E-29°E). In this longitude range, the intrinsic-to-total PSD ratio decreases with increasing time scales: above 50% in the submonthly range, 30-50% between 2 months and 2 years, and above 20% up to 5 years.

In the Ionian sea (15°E-20°E) the intrinsic-to-total PSD ratio exceeds 20% within the 5 months-3 year range. At 2°W in the Alboran Sea, this ratio exceeds 20% at timescales longer than 4 years. We recall that all the results presented in this study only concern the intrinsic and forced components of the non-seasonal variability that is generated within the Mediterranean Sea: besides the climatological boundary forcing in the Gulf of Cadiz, no other potential influence of the Atlantic is injected into our domain.

#### 5 Conclusion

This study explores from a 39-year 30-member ensemble 1/12° simulation the respective imprints of the forced variability and of the chaotic intrinsic variability (CIV) in the Mediterranean Sea. The first component is paced by the fluctuations of the external (atmospheric, river and lateral) forcing, and the second component spontaneously emerges from nonlinearities within the basin and has a random phase. We mainly focus on sea surface height (SSH) to diagnose the surface circulation and on the zonal transports within three layers associated with the density-coordinate overturning to evaluate the thermohaline circulation. We evaluate the relative contribution of intrinsic variability in these fields at high and low frequencies (periods

425

Figure 8. As figure 5 but for the intermediate transport variability (PSDs in color; units  $Sv^2.day$ ). The intrinsic-to-total PSD ratio is shown as 20%, 30%, and 50% isolines (thin solid, thick black, and red contours, respectively).

shorter and longer than about 2 years, respectively) using the metric  $R_i$ , the ratio between intrinsic and total variances, which varies between 0 (variability is fully forced) and 1 (variability is fully intrinsic).

Our analysis reveals that both high and low frequency SSH fluctuations are mostly random ( $R_i > 0.5$ ) over 17% of the Mediterranean sea, in particular in the Algerian Basin, the southern Ionian Sea, and the Levantine Basin. The Algerian Basin is the main hot spot of Mediterranean intrinsic variability, where up to 80% of the SSH variance is random at all periods between

435

4 months and 2 decades at least. Forced and intrinsic low-frequency variabilities have comparable amplitudes in the Rhodes, Bonifacio, and Alboran gyres; the relative contributions of both variability components have the strongest fluctuations in the latter gyre. In contrast, the simulated Northern Ionian Gyre (NIG) decadal fluctuations of the North Ionian Gyre are mostly paced by the atmosphere: less than 25% of its low-frequency variance is intrinsic and random. This implies that, in a realistic context with interannually varying air-sea fluxes, the timing of the observed reversals of the BiOS is likely imposed by the atmosphere.

Between Gibraltar and about 25°E, the variability of the upper overturning cell, which feeds the transport of Atlantic Waters (AW) to the east and of underlying intermediate waters to the west, is mostly paced by the atmosphere (and by the western boundary forcing at annual scale). However, an interannual-to-decadal intrinsic variability peak is found in the Alboran sea within this upper cell, where  $R_i$  reaches 30% for zonal transports in the surface and intermediate layers. East of 25°E,  $R_i$  increases only slightly in the surface transport, but reaches 35-40% for intermediate layer transports in the Levantine basin (26-30°E) at timescales shorter than 3 years. This peak is due to a large maximum of intrinsic variability in the lower overturning cell, which strongly randomizes bottom and intermediate zonal transports.

Forced and intrinsic fluctuations of bottom transports, and of intermediate transports to a lesser extent, also happen to peak at all time scales near convection sites, i.e. near 29°E (eastern Levantine Basin), 18°E (western Ionian Basin) and 6°E (Gulf of Lions). These longitudinal correspondences raise the hypothesis that forced wintertime convection and associated intrinsic variability may have a specific impact on the lower overturning cell there, in particular at 29°E where forced and intrinsic high-frequency variabilities have comparable influences ( $R_i \sim 30-50\%$  in intermediate and bottom transports). A preliminary search of potential covariances between deep convection and zonal transports, and between horizontal and overturning circulations, did not reveal any simple and clear connections. A more thorough investigation of such hypothetical links should focus on forced and intrinsic variability mechanisms separately, which is left for future studies.

Our results extend to basin-wide and multiple temporal scales earlier studies, which were focused on the Mediterranean intrinsic variability at seasonal time scales (Benincasa et al., 2024) and at interannual timescales in a specific region (Waldman et al., 2018). Chaotic intrinsic variability is substantial over most of the basin, reaches (decadal) periods that largely exceed those of mesoscale eddies, and competes with forced variability depending on regions and features of interest. However, intrinsic variance ratios in the Mediterranean sea tend to be smaller than in other parts of the global ocean (see e.g. Leroux et al., 2018; Carret et al., 2021), presumably since its dynamics are more constrained by the basin geometry and confined by its dimensions.

The spontaneous generation of multi-scale random fluctuations within the basin is a source of uncertainties that questions the attribution of regional fluctuations to sole atmospheric drivers, and that may hamper the predictability of its variability. Note that our experiment only gives access to the forced and intrinsic variabilities generated within the basin (as in most regional ensemble simulations, see e.g. Lin et al., 2023; Jamet et al., 2020), and restricts the Atlantic influence to its mean seasonal cycle within the western buffer zone. It would be interesting in future experiments to inject through the western buffer zone the forced and intrinsic variabilities coming from the Atlantic (using e.g. the global OCCIPUT ensemble run outputs), and to assess within the Mediterranean basin the relative contributions of endogenous and exogenous variability sources.

The numerical model configuration used in this study however shows certain deficiencies in representing emblematic Mediterranean Sea features and events, such as the Eastern Mediterranean Transient, the Ierapetra anticyclonic gyre, or the North Balearic cyclonic circulation. This prevented us from drawing conclusions about the impact of intrinsic variability on these features, encouraging further studies with improved model configurations.

The imprint of multi-scale CIV in several parts of the Mediterranean Sea also has practical implications on modeling and observing the basin. Indeed, even a perfect model would most likely differ from observations in terms of phase in the presence of CIV. On the one hand, this means that misrepresenting the phase of random events in an unassimilated single member simulation does not necessarily mean that the model is wrong. Ensemble simulations then provide modellers with an adequate means to explicitly represent and evaluate this stochasticity, and probabilistic assessment approaches that take all members into account (as in e.g. Narinc et al., 2024; Leroux et al., 2018) may be used to robustly assess the skill of numerical models in the presence of this CIV-related uncertainty. On the other hand, observationalists could favor strategies that capture as completely as possible the statistics of key phenomena, that is to say with a long-term repetition of measurements at the same location. This argues to favor observations from anchored platforms such as buoys or moorings, whose time series are less challenging to interpret in the presence of CIV than those captured by moving platforms such as ARGOs or gliders.

Author contributions. JMB designed and conducted the ensemble simulations using parameterizations provided by RP. TP and PB advised on methodological approaches and secured funding for the project. DH performed the diagnostics and, together with TP, led the writing of the manuscript. SS and RW contributed their expertise and assisted in the physical interpretation of the results. All contributed to the writing of the paper.

Competing interests. The contact author has declared that none of the authors has any competing interests.

Acknowledgements. We thank Pierre Nabat for developing and running the ALDERA3 forcing dataset and Florence Sevault for contributing to set and share the NEMOMED12 configuration and the associated forcings. NEMOMED12 and ALDERA3 are products developed and shared by the SiMED program. This study is a contribution to the Med-CORDEX international initiative (www.medcordex.eu). This work was undertaken as part of the MEDIATION project funded by the French government under grant agreement no. ANR-22-POCE-0003 of the France 2030 programme. This work was performed using HPC resources from GENCI-IDRIS (Grant 2024-A0140101279).

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
