# Peer review of "Externally-forced and intrinsic variability of the Mediterranean surface and overturning circulations"

_EGUsphere, 2025_

## Author Comment (AC1)

**ANSWER TO REVIEWER #2**

*We thank the reviewer for their careful reading of the manuscript and constructive comments. Their feedback has helped improve the clarity and presentation of the study. Our responses are shown in italic while the reviewer comments are in blue.*

The Authors use a 39-year, 30-member ensemble ocean simulation to assess basin-wide contributions of externally forced and intrinsic variability to key components of Mediterranean circulation across different scales.

Overall, the manuscript is well written, presents a relevant scientific topic, and employs appropriate methods. It is therefore suitable for acceptance after a few minor revisions suggested below.

- L33. Can you add one or more references where "chaotic intrinsic variability" (CIV) is first introduced/studied?

*To our knowledge, this expression has been first used in Bessières et al (2017) and Wolfe et al (2017). Both references have been added in the introduction.*

- L70. I suggest adding the approximate horizontal resolution in kilometers after „1/12°".

*Thank you for the suggestion, we have added the approximate horizontal resolution in kilometers after "1/12°" as requested.*

- L209. Please check the specified locations of the maximums and the corresponding potential densities in Figure 2a. The listed densities and the second location (24° E) do not appear to match the plotted values.

*Thank you very much for your careful reading. You are correct that the locations did not correspond to the maximum visible in Figure 2a. We have revised the manuscript accordingly.*

- L213 and L215. Typo „Straits" to „Strait".

*Corrected.*

- L355. I recommend rephrasing as: „Contrary to previous Mediterranean Sea multi-model evaluation studies (e.g., Dunić et al., 2019),..."

*In response to your comment, we have re-evaluated the relevance of comparing our results with a multi-model study, as it does not directly contribute to the discussion in our case. We now consider it more appropriate to focus on the agreement between our simulation and the observations, which is the key criterion for this study, and have therefore decided to remove the reference to Dunic et al. (2019).*

- L357-360 and L428-431. Could you elaborate further on the conclusion that the atmosphere is the predominant external driver of decadal fluctuations of the NIG (i.e., BiOS)?

This is an interesting question. Our ensemble run indeed indicates that decadal NIG fluctuations mostly belong to the ensemble mean ocean response, i.e. that they are mostly forced externally. The atmosphere is the only external driver that can force interannual variability in our simulation since all other external drivers (western boundary forcing and runoffs) are devoid of interannual fluctuations. We realize that we did not comment much on the mostly forced character of NIG variability in the former version of the conclusion, that we have complemented with the following lines : "*This result contrasts with some previous studies based on observational data, idealized simulations, and laboratory experiments, which*

*highlighted the potentially dominant role of CIV in NIG fluctuations (see introduction). Our simulation confirms that CIV has an impact on the NIG behavior, but suggests, as do other realistic ocean modeling studies, that taking into account complex stratification, basin geometry, and fully variable atmospheric variability gives more weight to external factors than to intrinsic processes in the timing of these fluctuations. Future studies will help further assess the robustness of these conclusions."*

- L434 and L451. Typo „sea" to „Sea".

*Thanks for mentioning this error. Corrected.*

- I suggest adding a short discussion on how horizontal resolution may influence the intrinsic variability.

*Thanks for suggesting (as did the other reviewer): this is now done in the conclusion.*

---

## Author Comment (AC2)

**ANSWER TO REVIEWER #1**

*We thank the reviewer for mentioning the robustness and relevance of the study. We have carefully addressed all the comments below. Our responses are shown in italic while the reviewer comments are in blue.*

Héron et al. analyse a 39-year, 30-member, 1/12° ensemble simulation of the Mediterranean Sea to distinguish intrinsic from forced variability in surface and zonal overturning circulation. Ensemble statistics and temporal scale decomposition reveal that SSH variability is predominantly intrinsic over about 17% of the basin at both timescales, with hotspots in the Algerian, Levantine, and Ionian Seas. Zonal overturning variability is largely atmosphere-driven, though intrinsic processes remain important in intermediate and deep layers near convection sites, particularly in the Levantine basin. Using relative vorticity, the authors further show that Mediterranean gyres span a continuum from strongly forced (e.g., North Ionian Gyre) to largely intrinsic (e.g., Algerian Gyre).

The manuscript is clearly written, with some minor structural aspects that could be refined, and supported by robust results. It addresses a relevant scientific question within the broader context of detection and attribution. Although the study is primarily descriptive, it targets a region where such characterization is still limited and provides insights that can meaningfully advance our understanding of Mediterranean Sea dynamics. I recommend acceptance with the following minor revisions:

- I would suggest considering a revision of the abstract, as it currently reads a bit convoluted and may not clearly convey the main results of the study.

*Thanks for this suggestion: we have tried to clarify the wording of the abstract. While we think that our main results are summarized in the abstract, we have slightly refined the description of intrinsic variability in the Levantine Basin:*

> *Original: "in particular in the Levantine basin where this fraction exceeds 50 % between 27 and 30° E at submonthly periods, and 20–30 % at periods reaching 20 years locally."*

> *Revised: "in particular in the Levantine basin where this fraction exceeds 50% between 27 and 30°E at submonthly periods, 30% at subannual periods, and 20% at longer periods (reaching 20 years locally)."*

*After these corrections, it seems to us that the abstract provides a quite exhaustive summary of the study.*

- L74 - 83: The manuscript provides an extensive description of the ALDERA3 dataset. As I understand, this dataset was not produced by the authors but obtained from another source. I recommend clarifying this point explicitly. In that case, the description could be shortened to include only the essential aspects relevant to the study, with a reference to the original documentation for further details.

*The ALDERA3 forcing dataset has indeed been produced by the team of authors (with Pierre Nabat, who is acknowledged in the dedicated section). This dataset is used for the first time to force an ocean simulation; our paper might be cited as a reference for future studies using ALDERA3. For the latter two reasons, we chose to describe this dataset. In order to underline the novelty of this ALDERA version, we added the adjective "new" in the first sentence of this paragraph ("Our surface forcing is provided by the new ALDERA3 dataset…").*

- Subsection 2.3.3: Could you add a comment on why you chose 2.5 years as a threshold to distinguish the LF and HF components of the timeseries?

*Let us first mention that choosing a timescale to split a wide frequency spectrum in two bands is arbitrary and not constrained by a specific need; different authors might make different choices that are equally valid. Here are the reasons for our choice: after testing other threshold timescales, we found that the middle and bottom rows of Figures 1 and 4 are well balanced with this choice of 2.5 years: HF and LF SSH variability maps share the same colorscales and are easy to compare. In other words, this choice of 2.5 years splits the total, forced and intrinsic variances of SSH into high- and low-frequency reservoirs that have comparable magnitudes. It also turns out that with this choice, both Ri maps also share common features, and are clearer. Since this choice of timescale is inherently arbitrary (yet legitimate in our opinion) rather than being constrained by any statistical or oceanographic reason, we suggest not to address it in the paper.*

- L124: If I understood your equation correctly, I think it should be H(σ0(x,y,z,t,n) - σ0).

*Thanks for mentioning this error. Corrected.*

- L179 - 185: I would comment differently on the HF SSH variability. The model is in good agreement with the observation and it is indeed underestimating the variability along the Algerian coast and the Levantine basin. However, this is not true in the north-western part of the basin, in the Northern Adriatic Sea, and in the Gulf of Gabes.

*Indeed, the simulated SSH HF total variability tends to exceed this observational estimate very close to the coasts of the Gulf of Lion and of the Gulf of Gabès (but this is not as clear along the northern Adriatic coast). However, the width of these 2 coastal maxima does not exceed 100 km, a scale that is resolved by the model but smaller than the effective resolution of DUACS altimetric products at mid latitudes along the coasts (coarser than 120km, see i.e. Fig 14a in [https://doi.org/10.5194/essd-15-295-2023](https://doi.org/10.5194/essd-15-295-2023)). It is therefore very unlikely that DUACS captures these simulated coastal HF variability maxima; we now mention in this section these simulated maxima: "On the opposite, HF variability maxima are simulated in coastal bands narrower than 100 km in the Gulf of Lion and of the Gulf of Gabès, which do not appear as clearly in the altimetric estimate whose effective resolution is likely too coarse. We will come back to the probable origin of these narrow bands in the following."*

*In order to slightly clarify this section 3.1.2, we now start the second sentence of this section by "In these regions".*

- Figure 1: I think that the discussion about the specific gyres would be more easily understandable if the boxes shown in Figure 1 were named or referred to in some way in the Figure itself or in the caption. You could use numbers, letters, or acronyms to refer to each specific box. Moreover, I would recommend changing the color of the axes of the left plots to something other than yellow and that of the boxes in panels e and f to something brighter.

*Thanks to your suggestion: we have adjusted the axis color, which was not very readable, and added references for the boxes: Eastern Alboran Gyre, Algerian Gyre, Bonifacio Gyre, North Ionian Gyre (NIG), and Rhodes Gyre.*

- In subsection 3.2.2, I would add that the bottom zonal transport at low frequency is comparable to the surface and intermediate zonal transports in the regions 17 - 21°E and 27 - 30°E, and comment on that.

*Thanks for this interesting remark that is worth mentioning indeed. We have added the following comment at the end of section 3.2.2: "These two longitude ranges are also those where low-frequency zonal transport fluctuations are weaker at the surface than at the bottom: intermediate zonal transports are thus mostly modulated there by the lower ZOC cell at low frequency."*

*Note that we have also slightly clarified the sentence that follows and that concludes the section.*

- The horizontal lines in panels c and f of Figure 4 are defined later in the text, in section 4.1.3. I would suggest adding that to the Figure's caption as well.

*Thank you for the suggestion. We have taken your comment into account and added this information to the caption of Figure 4.*

- L275 - 276: Maybe add a short comment on why large values of forced HF SSH variability are found in the Alboran Sea and in the Tunisian shelf.

*Thanks for this suggestion, which however made us realize that attributing each of these forced HF SSH variability maxima to a specific driver (and thus answering your specific question) cannot be done properly without dedicated sensitivity ensemble experiments. We thus rewrote this paragraph in order to clarify it, now mentioning too potential promoters of forced HF variability in the 5 reported maxima: the concave and shallow character of most concerned areas, and the strong variability of locally dominant winds (including in the Alboran Sea).*

- I would unify subsections 4.1.1 and 4.1.2 into a single one since a comparison between HF and LF variability is made, focusing on the similarities and the differences between the two temporal scales. Could you give an interpretation of the reason why the two Ri maps generally agree except for the Tyrrhenian Sea and the Ionian Sea?

*- Unification :*

*Subsections 4.1.1 and 4.2.2 (as well as 4.3.1 and 4.3.2) were unified in a former version of the paper, but we preferred to split them as is for 3 reasons:*

*[1] A merged section would have to discuss the 6 subplots of Figure 4 and their mutual relations, which might make the discussion (and reading) confusing to the reader.*

*[2] the fact that CIV may be locally large (or dominant) at HF does not come as a true novelty, since mesoscale turbulence is a well-known source of HF randomness. Section 4.1.1 mostly discusses this.*

*[3] That LF variability is roughly as random as its HF counterpart is newer in the Med Sea, and we wanted to dedicate a subsection (4.1.2) to this.*

*Actually, the spectral analysis in section 4.1.3 provides a unification between both ranges of timescales afterwards.*

*- Interpretation : local disagreements between Ri maps may be due to many factors. Forced and intrinsic variabilities (and thus Ri) at both timescales presumably depend on oceanic/atmospheric features that are very likely inhomogeneous; disagreements between Ri maps are thus not so surprising. In our opininon, what is more surprising and perhaps striking is the general agreement between both Ri maps (this agreement is also a benefit of the choice of a 2.5 year separation timescale). We have no clear explanation for this yet, but this result conveys in a simple way that in most regions, the random fraction of SSH variance is as large on both sides of the separation timescale at first order.*

*However, explaining in detail the general agreement between maps, and these 2 local disagreements remains difficult at this stage, and would require more dedicated studies.*

- I would suggest doing the same for subsections 4.3.1 and 4.3.2. Could you also further comment on the different results obtained for HF and LF variance of zonal transport, especially about the difference in magnitude between the two?

*If possible, we would prefer to separate the discussion about HF and LF variances of zonal transports as we did for SSH, mostly to facilitate the description and discussion of the 6 subplots of Figure 7.*

*We must acknowledge that we do not have a definitive explanation for the larger STDs of zonal transports at HF than at LF. However, knowing that deep water formation evolves on short time scales (down to a few days or weeks: Waldman et al., 2017; Testor et al., 2016) and directly influences the overturning (e.g Waldman et al., 2018), we may expect that the part of the transports' fluctuations that is sensitive to deep convection also has a strong subannual component (which is captured in our HF band). Providing a more robust answer to this interesting question would require a dedicated investigation, which we leave for the future.*

- I would recommend stressing more the relevance of the analysis of individual gyres and the motivation behind it.

*Thanks for this suggestion. The analysis of the NIG is mostly motivated by the active debate about the intrinsic vs extrinsic sources of this gyre's interannual variability (as we had mentioned —perhaps too discretely— in section 4.2 in the first draft); the analysis of the four other gyres (that are often studied in the basin as mentioned in the second paragraph of the introduction) is motivated by the will to assess the potential diversity of gyre dynamics across the basin. We do agree that these motivations (in particular the imprint of CIV on the NIG) were introduced too late in the former version of the paper: we now explicitly mention them in the introduction, and recall them in section 4.2. Both parts have been rephrased to put these motivations forward: thank you for this suggestion.*

- Have you checked how dependent your results are on the ensemble size? Please comment on that.

*There is unfortunately no simple answer to this good question. Increasing the size N of ensemble simulations enhances statistical accuracy (depending on the quantities of interest: https://doi.org/10.5194/esd-11-885-2020) but increases computational costs: choosing N is thus a compromise. N typically ranges from a few to a few tens in geosciences (6 to 100 in https://doi.org/10.1016/j.jhydrol.2024.131679); in this context, N=30 is a usual, relatively large yet affordable choice that we also made, although more members (as well as increased resolution) would yield even more accurate results(\*).*

*(\*): Note that one of our former studies about the forced and intrinsic AMOC variability (https://doi.org/10.1175/JCLI-D-17-0168.1), based on a 50-member global ensemble, quantitatively confirmed the results formerly estimated from a minimal (2-member) ensemble performed with a similar model (https://doi.org/10.5194/os-9-805-2013). In that case, the 25-times increase in ensemble size obviously enhanced the accuracy and robustness of these earlier estimates, but showed that a much smaller ensemble had some quantitative skill in estimating both variability components. This suggests that our standard but relatively large (30-member) ensemble is likely suitable for the present study.*

- In the conclusion, among the possible deficiencies of the model used, I would emphasize the well-known dependence of internal variability on horizontal resolution. The model has a resolution of 1/12°, which, in the Mediterranean Sea, is often much larger than the Rossby radius of deformation.

*Thanks for these suggestions.*

*- Grid step vs Rossby radius. Fig 4 in https://rjes.wdcb.ru/v20/2020ES000737/2020ES000737.pdf shows that the model grid step (7-8 km) is not exactly "much larger", but close to or even finer than the Rossby radius, except in small shallow areas and in the Adriatic Sea. Away from these regions, this provides our model a so-called "eddy-permitting" character, comparable to that of a global 1/4° model up to about 40-45° latitude where both the Rossby radius and the grid step are close to 20-30km (Chelton et al, 1998). The ability of our 1/12° ensemble to simulate dynamical instabilities (and subsequent CIV) in the Mediterranean Sea is thus comparable to the ability of the 1/4° global OCCIPUT ensemble to simulate them at mid-latitudes. It is true however that a resolution finer than 1/12° would be needed to make the*

*simulation more "eddy-resolving".*

*- Dependence of CIV on resolution. It is indeed "well-known" that finer model resolution tends to enhance mesoscale (i.e. HF) CIV. However, how a resolution finer than 1/12° may impact the CIV of on our main variables of interest (SLA and overturning) over HF and LF ranges is not as clear: switching from 1/4° to 1/12° resolution indeed enhances the LF CIV of sea-level (Fig 10 in [https://doi.org/10.1175/JCLI-D-14-00554.1](https://doi.org/10.1175/JCLI-D-14-00554.1)), but barely affects the LF CIV of the Atlantic overturning (Fig 7 in [https://doi.org/10.1175/JPO-D-14-0163.1](https://doi.org/10.1175/JPO-D-14-0163.1)).*

*We nevertheless agree with the reviewer that the conclusion should mention the potential sensitivity of our results to model parameters and resolution. We have summarized the above discussion in the conclusion as follows: "Whether intrinsic variability would further increase at finer horizontal resolution is very likely at mesoscale, but not certain in general: tripling the resolution in a global simulation did not significantly increase the Atlantic overturning interannual CIV \citep{gregorio_intrinsic_2015}. Assessing the robustness of the present results to model resolution and other parameters is left for future studies."*

- Even though you have highlighted the differences in the conducted analysis compared to the previous studies that you cited (Benincasa et al., 2024, Waldman et al., 2018, and the OCCIPUT project), it would be valuable to include a concise but explicit comparison of your findings with the results reported in those works.

*Thanks for your valuable suggestion: we have revised the conclusion to better contextualize our results relative to Benincasa et al. (2024), Waldman et al. (2018), and OCCIPUT studies. More precisely:*

*As now stated more clearly in the introduction and in the conclusion, Benincasa et al (2024) have described certain impacts of forced and intrinsic variability throughout the basin, but only at seasonal timescale, which we removed from our timeseries by deseasonalization. Our work thus complements their study by providing results over a range of timescales distinct from seasonal: comparison between our and their results is thus not possible, and is not included in the conclusion.*

*Waldman et al (2018) have characterized the imprint of forced and intrinsic variabilities from daily to interannual timescales, but only on deep convection in the Gulf of Lions; we focus on different variables (SSH, ZOC) at the scale of the basin: comparison with their results is thus difficult to make as well. However, they report a substantial impact of CIV on deep convection volume in the Gulf of Lions from daily to interannual timescales; this finding may be consistent with ours in the same region (and in other convection sites in our case), since a spread in the volume of newly formed deep water may drive a spread in its deep and intermediate advection via ZOC-related flows (which appears in our Figs 7b,c,e,f). However, given the distance between both studies and the relative complexity of this hypothesis, we prefer not to formulate it too explicitly in the conclusion; we instead mention a perspective aiming to further examine potential connections between the imprints of forced and intrinsic variabilities on deep convection on the one hand, and ZOC-related transports on the other hand.*

*Analyses of the 1/4° OCCIPUT ensemble have shown substantial imprints of low-frequency CIV on SSH at global scale (e.g. Carret et al 2021), and on the (meridional) overturning in the Atlantic (Leroux et al 2018). The present study does confirm in the Mediterranean Sea the existence of CIV impacts on SSH, and on the (zonal) overturning. Although Carret et al did not focus on the exact same range of timescales as we did, their Fig 3b suggests that sigma_i for SSH is 2cm smaller than our estimates (our Fig 4e); we attribute this difference to model resolution (1/4° vs 1/12°), as was shown by Serazin et al (2015). These considerations are now summarized in the conclusion.*

Technical corrections:

- L6: I suggest writing ".. 75% of the interannual to decadal SSH variance of the North Ionian Gyre circulation ..".

*Corrected.*

- L12: I suggest rephrasing "and 20-30% at periods reaching 20 years locally" to make it clearer.

*Following your suggestion about the abstract we have slightly refined the description of intrinsic variability in the Levantine Basin ( see our answer to reviewer #1's comment about the abstract).*

- The multiplication symbol in the units of several plotted quantities and throughout the text is not rendered well in the document. I would suggest double-checking the LaTeX expression.

*Yes indeed. We have corrected the multiplication symbol where necessary.*

- L76: Missing parentheses for references "Nabat et al., 2020" and "Colin et al., 2010" on the next line.

*Yes indeed: corrected.*

- L253 - 259: It seems like the same concept, i.e. that the total HF variability of the intermediate westward transport is weaker than that of the eastward surface transport west of 10°E, is repeated two times.

*Thank you for pointing this out. We clarified the text to avoid the misleading repetition. We now explicitly state that the coherence between surface, intermediate and bottom high-frequency transport anomalies is what supports the interpretation regarding the sign of the upper and lower ZOC cells.*

- L270 - 271: I would position the introductory sentence "The left .. at LF (bottom)" in section 4.1 before the beginning of subsection 4.1.1.

*We have taken your comment into account.*

- L290: typo in "Figure4d".

*Corrected.*

- Figure 5: Specify that you are showing the log of the Power Spectral Density in the caption.

*Thank you for your careful reading; we have taken your comment into account.*

- Figure 6: Increase the size of the title of panel b.

*Corrected.*

- L339: typo $R_i(t)$.

*Corrected.*

- L366: typo "behaviors".

*Corrected.*

- Regarding the multi-scale character of the Mediterranean Sea, I would suggest adding the following reference: Robinson, A. R., Leslie, W. G., Theocharis, A., and Lascaratos, A.: Mediterranean sea circulation, Ocean currents, 2001, 1689– 1705,

https://doi.org/10.1006/rwos.2001.0376, 2001.

*Thank you for this relevant suggestion. We have added the recommended reference to the manuscript.*